# The computational relationship between reinforcement learning, social inference, and paranoia

**Joseph M. Barnby**[1,2,3]*, **Mitul A. Mehta**[2,3], **Michael Moutoussis**[4,5]

1 Department of Psychology, Royal Holloway, University of London, London, United Kingdom, 2 Cultural and Social Neuroscience Group, Department of Neuroimaging, Institute of Psychiatry, Psychology & Neuroscience, King's College London, University of London, London, United Kingdom, 3 Neuropharmacology Group, Department of Neuroimaging, Institute of Psychiatry, Psychology & Neuroscience, King's College London, University of London, London, United Kingdom, 4 Wellcome Centre for Human Neuroimaging, University College London, London, United Kingdom, 5 Max-Planck–UCL Centre for Computational Psychiatry and Ageing, University College London, London, United Kingdom

* joseph.barnby@rhul.ac.uk

**Data Availability Statement:** All data, models and analysis code are available on Github (https://github.com/josephmbarnby/Barnby_etal_2022_ReversalLearning/).

## Abstract

Theoretical accounts suggest heightened uncertainty about the state of the world underpin aberrant belief updates, which in turn increase the risk of developing a persecutory delusion. However, this raises the question as to how an agent's uncertainty may relate to the precise phenomenology of paranoia, as opposed to other qualitatively different forms of belief. We tested whether the same population (n = 693) responded similarly to non-social and social contingency changes in a probabilistic reversal learning task and a modified repeated reversal Dictator game, and the impact of paranoia on both. We fitted computational models that included closely related parameters that quantified the rigidity across contingency reversals and the uncertainty about the environment/partner. Consistent with prior work we show that paranoia was associated with uncertainty around a partner's behavioural policy and rigidity in harmful intent attributions in the social task. In the non-social task we found that pre-existing paranoia was associated with larger decision temperatures and commitment to suboptimal cards. We show relationships between decision temperature in the non-social task and priors over harmful intent attributions and uncertainty over beliefs about partners in the social task. Our results converge across both classes of model, suggesting paranoia is associated with a general uncertainty over the state of the world (and agents within it) that takes longer to resolve, although we demonstrate that this uncertainty is expressed asymmetrically in social contexts. Our model and data allow the representation of sociocognitive mechanisms that explain persecutory delusions and provide testable, phenomenologically relevant predictions for causal experiments.

## Author summary

Responding to shifts in inanimate and social environments is important for adaptation and appropriate communication. Studies have demonstrated generic cognitive distortions

**Funding:** JMB was supported by the UK Medical Research Council (MR/N013700/1) and King's College London member of the MRC Doctoral Training Partnership in Biomedical Sciences. MM is supported by the Wellcome Trust as a member of the 'Neuroscience in Psychiatry Project' (NSPN) which is funded by a Wellcome Strategic Award (ref 095844/7/11/Z). The Max Planck – UCL Centre for Computational Psychiatry and Ageing is a joint initiative of the Max Planck Society and UCL. The funders had no role in study design, data collection and analysis, decision to publish, or preparation of the manuscript.

**Competing interests:** The authors have declared that no competing interests exist.

to the processing of information in shifting contexts to underpin or accompany the development of symptoms of severe mental disorders, such as persecutory delusions. However, given the clear social phenomenology and clinical needs regarding social function which accompany persecutory delusions, explanations that detail how changes in generic cognition dovetail with social cognition are urgently needed. We addressed this gap by measuring the relationship between computational mechanisms governing non-social decision making and social inferences upon reversal of task contingencies, and the impact of pre-existing paranoia. We found that paranoia was related to uncertainty in both non-social and social contexts, and crucially, increased non-social uncertainty was related to changes in sociocognitive parameters. Paranoia was related to context-dependent, asymmetric biases in prior beliefs and belief-updating in social contexts. Importantly, paranoia increased the propensity to explain behaviour shifting away from beliefs about harm intent through alternative attributions. Our model and data bridges non-social and social theory explaining persecutory delusions and provides a mechanistic, phenomenologically relevant framework for causal experiments.

## Introduction

The ability to make inferences about the environment when it changes is crucial to survival and adaptation. This is especially important when interacting with other people, where recognising and interpreting violations of our predictions is crucial for communication, cooperation and taking defensive action.

Psychiatric disorders are characterised by difficulties in social interaction and poor adaptation to new environments. In the case of persecutory delusions, individuals hold unwarranted beliefs that others intend to harm them, even in the absence of tangible evidence. Formal modelling of choice behaviour has suggested paranoia is characterised by increased perseveration and greater non-deterministic action preferences which are attributed to higher expectations of volatility in the environment [1–4]. These studies used probabilistic learning tasks with changing reward probabilities over time, in the absence of a discernible agent controlling the contingency shifts (e.g., [5–6]). To examine reinforcement learning observations within social contexts relevant to paranoia, experimenters have also framed probabilistic tasks in terms of interaction with social agents, demonstrating that those with higher paranoia are slower learners and more sensitive to changes in the social environment [7], more rigid in their beliefs about partners [8], and less likely to take advice from partners [9–10].

Experimentally demonstrating the phenomenological relevance of reinforcement learning in paranoia is important as we move as a field to develop more precise formal models of persecutory delusions. Current neurocognitive theories of persecutory delusions suggest associative learning mechanisms underpin the development of positive symptoms in psychosis [11–12], particularly through poor integration of lower perceptual information leading to uncertainty over beliefs about the world [13]. However, theories that implicate the role of reinforcement learning biases in persecutory delusions need to explain *how* learning biases lead to phenomenologically relevant experiences that form the basis for current cognitive models of persecutory delusion formation and maintenance in the clinic [14–16]. Indeed, the necessity to build formalised model which can accommodate the rich state space of social contexts have been called for more broadly [17]; formal explanations of social interaction must ensure learning is outlined explicitly in relation to how we probabilistically represent beliefs about ourselves and others.

In this set of experiments, we build bridges between formal, domain-general accounts of probabilistic reasoning and changes to social-cognitive representations central to paranoia. We tested whether participants varying in paranoid ideation displayed differences and/or commonalities in social and non-social reversal learning, inference, and decision consistency. If paranoia is simply an example of a dysfunctional but general reinforcement learning mechanism applied to social interaction, we should expect all types of motivational attributions to be influenced in similar ways, irrespective of content: harmful intent and self-interest judgements should both be affected in parallel by higher pre-existing paranoid beliefs when changes in a partner's behaviour could be due to either motive. Alternatively, if intention attributions are not affected in the same way by a partner's behavioural changes, it is likely that domain-general neurocomputational changes are subject to differentiated interactions with the specifics of social cognition. This makes it important to understand the mechanisms giving rise to social asymmetries. We used conceptually similar probabilistic social and non-social tasks in the same large population to detect such key cognitive differences. Building on previous work [18], we built separate computational models to capture behavioural (choice) and inferential differences within each task. Each model quantified decision/inferential uncertainty as precision in the agent's decision making, or precision of an agent's beliefs about how closely their partner's decisions reflected their true intent, respectively. Each model also quantified participants' response to contingency reversals.

In line with prior evidence, we predicted that during the probabilistic reversal learning task paranoia would be associated with lower decision consistency, greater win-switch rates, and greater perseveration errors following the reversal. In the modified repeated reversal Dictator game, we hypothesised that higher paranoia would lead to rigidity in harmful intent attributions formed about a partner when a partner's behaviour changes, regardless of whether they were fair or unfair pre-reversal. In an exploratory analysis we tested the relationship of individual parameter values in the non-social task with parameters derived from the social model to understand how biases in probabilistic learning may be expressed in social contexts.

## Results

We administered a non-social probabilistic reversal learning task and a modified repeated reversal Dictator Game to 693 participants, in addition to collecting data on participants persecutory ideation (hereafter termed 'paranoia'; measured via subscale B of the Revised Green Paranoid Thoughts Scale; R-GPTS [19]), general cognitive ability (using the International Cognitive Ability Resource–Progressive Matrices {ICAR} [20]), age, sex, and task comprehension. We conducted computational model-agnostic and model-based analyses; in model-based analyses, we tested a range of associative models for the non-social task (k = 8), and a range of associative (k = 7) and Bayesian-belief (k = 6) models in the social task to account for participant choice and attributional behaviour, respectively. In addition to reporting model-based and model-agnostic outcomes for each paradigm, we report the relationship between key parameters across winning non-social and social computational models (see Fig 1 and Methods for more details).

R-GPTS scores were highly skewed to the left and low (mean [sd] = 3.88 [6.18], skew = 2.22, range = [0, 33]). Compared to previously reported norms on the R-GPTS subscale B (mean = 2.53; [19]), our population had significantly higher scores on average ($t_{(692)}$ = 5.72, p < 0.001), but lower than the typically reported cut-off clinical mean (mean discriminatory of clinical populations = 11; $t_{(692)}$ = -30.29, p < 0.001). ICAR scores were normally distributed (mean [sd] = 4.96 [2.42], skew = 0.08) and not significantly different to previously reported means ([20]; mean = 4.97; $t_{(692)}$ = -0.16, p = 0.87).

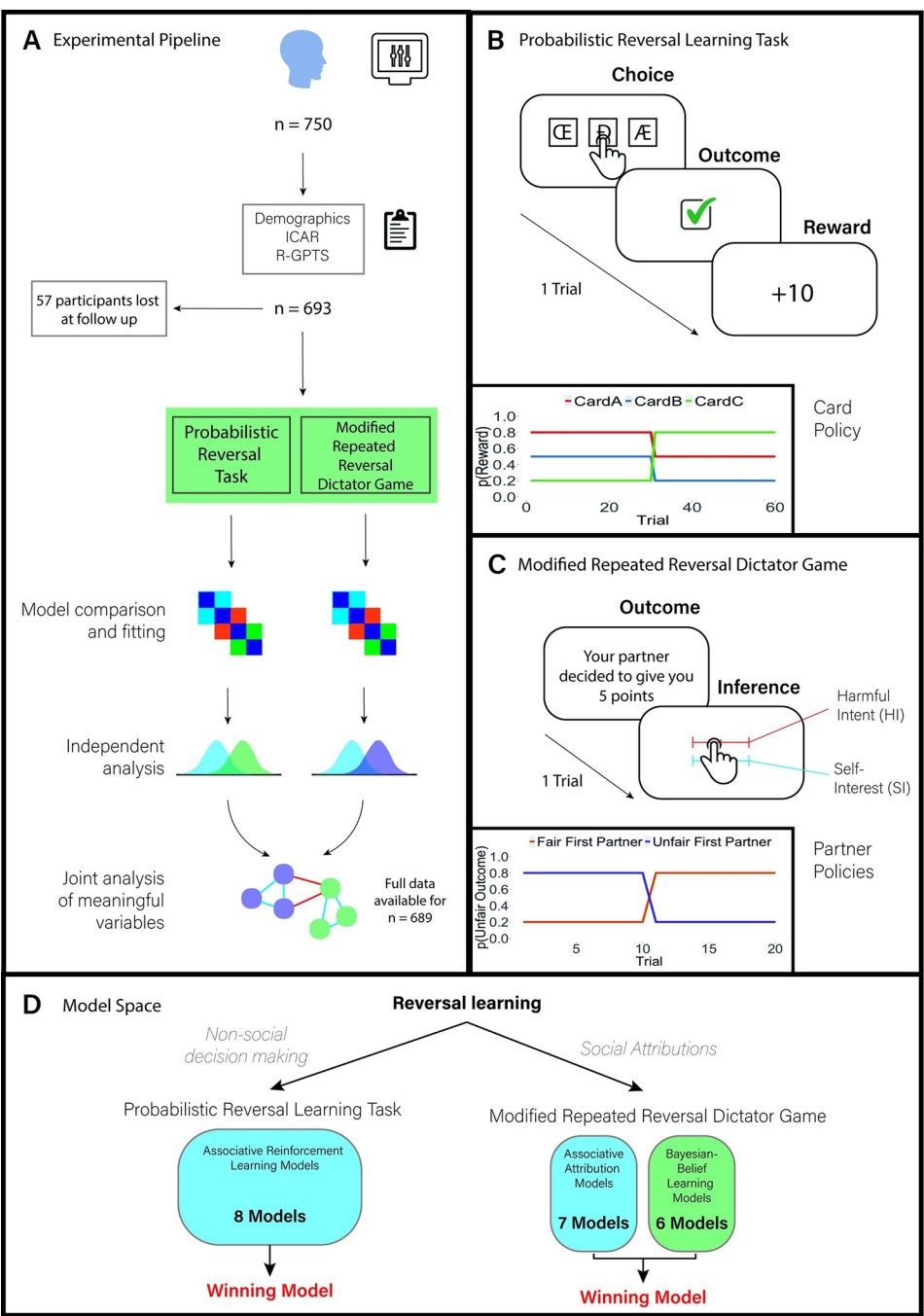

**Fig 1. Study design.** (A) Experimental design and analysis plan for each paradigm. (B) An example of a trial from the probabilistic reversal paradigm. There were 60 trials in total, and after 30 trials, the contingency of the rewarding card changed unknown to the participant. (C) Example trial from the modified repeated reversal Dictator Game, where participants had to infer their partner's intent. There were 20 trials in total, and after 10 trials, the contingency of the Dictator changed unknown to the participant. Participants were paired with a partner who was either at first more likely to be fair or unfair, and then changed their policy after the reversal. (D) Model space. Reversal learning was assessed across both non-social decision making and social attributions, using a probabilistic reversal learning task and modified repeated reversal Dictator game as measurement tools, respectively. All models were assessed using MAP estimation with weak priors. The winning models across both Bayesian-belief and associative classes within the repeated reversal Dictator Game were further assessed using Concurrent Bayesian Modelling (Piray et al., 2019).

## Computational model-agnostic analysis

**Probabilistic reversal learning task.**   In sum, after controlling for confounders, paranoia was positively associated with choosing the worst card following a reversal. Paranoia was only associated with earning fewer rewards and win-switch biases following reversals. Paranoia was not associated with less accurate forced-choice self-reports asking which was the best card.

We first report raw associations between paranoia and cognition, and then account for key covariates, as per pre-registration. Paranoia was not associated with the trial-by-trial probability of choosing the optimal card (80/20 card) before the reversal (-0.01, 95%CI: -0.06, 0.11), but was after the reversal (-0.12, 95%CI: -0.22, -0.02; S1 Fig). The worst card (with a 20/80 chance of reward) was chosen significantly more on a trial-by-trial basis in those with higher paranoia after the reversal (0.06, 95%CI: 0.02, 0.09; S2 Fig), but there was no relationship between paranoia and the probability of choosing the card with 50/50 probability of reward after reversals. Paranoia was not associated with fewer rewards prior to reversal (0.05, 95%CI: -0.02, 0.13) but was after reversal (-0.12, 95%CI: -0.20, -0.05). Paranoia was associated with win-switch rates after reversals (the probability that after receiving a reward, participants selected a different card on the next turn; 0.12, 95%CI: 0.05, 0.19) and lower lose-stay rates after reversal (after not receiving a reward, participants stick with the card they last selected; -0.08, 95%CI: -0.15, -0.00). Calculating rates across all trials as previously analysed [21] showed paranoia was associated with win-switch rates (0.10, 95%CI: 0.03, 0.17) but not lose-stay rates (-0.05, 95%CI: -0.12, 0.02). Finally, when participants self-reported which card gave the most rewards at the end of the task, paranoia was not associated with fewer correct answers before the reversal (0.00, 95%CI: -0.03, 0.03), nor after reversal (-0.02, 95%CI: -0.05, 0.01)

When we adjusted for age, sex, ICAR score, and task comprehension, the remaining associations with paranoia were the relationships with fewer optimal card selections (-0.08, 95%CI: -0.20, -0.00; see online code supplement; regression model P2; S1 Fig), selections of the worst card after the reversal (0.04, 95%CI: 0.01, 0.08; model P2b), greater rewards prior to reversal (0.08, 95%CI: 0.01, 0.16; model P4a), fewer rewards after reversal (-0.11, 95%CI: -0.18, -0.03; model P4b), and larger win-switch rates after reversal (0.09, 95%CI: 0.02, 0.17; model P5a).

Accounting for covariates abolished win-switch rates across all trials (0.06, 95%CI: -0.01, 0.13; model P5a), as well as lose-stay associations after reversal (-0.06, 95%CI: -0.14, 0.02; model P5b). Paranoia was still not associated with the probability of choosing the optimal card before the reversal (0.03, 95%CI: -0.06, 0.11; model P1), nor with lose-stay rates (-0.01, 95%CI: -0.09, 0.04; model P5b), and nor with fewer self-reported correct answers before the reversal (0.04, 95%CI: -0.15, 0.24; model P4a) or after the reversal (-0.01, 95%CI: -0.29, 0.11; model P4b).

ICAR scores were associated with both lower win-switch (-0.15, 95%CI: -0.22, -0.08; model P5a) and greater lose-stay rates (0.19, 95%CI: 0.12, 0.26; model P5b) across all trials in the same adjusted models where it was included as a covariate. In exploratory analysis we also allowed paranoia and ICAR scores to interact in separate auxiliary models. Paranoia and ICAR scores did not interact to predict win-switch rates (0.04, 95%CI: -0.01, 0.15; model P5a-Aux), nor interacted to predict lose-stay rates across all trials (interaction not included in final top model; model P5b-Aux).

**Modified repeated reversal dictator game.**   In brief, after controlling for confounders, paranoia was associated with larger and less flexible harmful intent attributions (HI). Paranoia did not influence self-interest attributions (SI).

Again, we first report raw associations with paranoia, and then account for key covariates. Across all trials there was an influence of initial partner behaviour on HI (0.44, 95%CI: 0.32,

0.55) and SI (0.81, 95%CI: 0.71, 0.91), such that initially unfair partners were associated with greater HI and SI. There was also an interaction between initial partner behaviour and attributions before and after the reversal (HI: -0.93, 95%C: -0.98, -0.89; SI: -1.20, 95%CI: -1.25, -1.15), such that both HI and SI less after an initially unfair dictator became fair, compared to when an initially fair dictator became unfair. Paranoia was associated with HI (0.12, 95%CI: 0.06, 0.17), but not SI (-0.03, 95%CI: -0.07, 0.02) across all trials. Paranoia interacted with reversals, such that HI changed less after reversal as paranoia increased (-0.05, 95%CI: -0.08, -0.03). There was no interaction between paranoia and trials after reversal concerning SI (-0.01, 95% CI: -0.04, 0.02).

We then examined adjusted effects. There was an influence of initial partner behaviour on both attributions, with partners who were initially more unfair inducing higher attributions compared to partners who were initially fairer (HI: 0.43, 95%CI: 0.31, 0.55; model S1a; SI: 0.82, 95%CI: 0.72, 0.91; model S1b). There was still also an interaction between initial partner behaviour and attributions before and after the reversal (HI: -0.93, 95%C: -0.98, -0.89; SI: -1.20, 95%CI: -1.25, -1.15), such that both HI and SI changed less after an initially unfair dictator became fair, compared to when an initially fair dictator became unfair. Paranoia was associated with higher HI (0.10, 95%CI: 0.04, 0.16; model S1a) but not SI (-0.01, 95%CI: -0.07, 0.03; model S1b) across the board. Paranoia interacted with reversals, such that HI changed less after reversal as paranoia increased (-0.05, 95%CI: -0.08, -0.03). There was no interaction between paranoia and trials after reversal for SI (-0.02, 95%CI: -0.07, 0.03). We additionally allowed paranoia and initial partner behaviour to interact. There was no meaningful interaction between paranoia and initial partner behaviour for either attribution (HI: 0.07, 95%CI: -0.04, 0.18; model S3a; SI: -0.01, 95%CI: -0.07, 0.03; model S2b).

ICAR scores were associated with lower HI (-0.14, 95%CI: -0.20, -0.09; model S1a) but not SI (model S1b). In exploratory auxiliary models, we allowed paranoia and ICAR scores to interact, although this interaction was not associated with HI (0.01, 95%CI: -0.03, 0.10) nor SI (0.01, 95%CI: -0.01, 0.10).

## Computational model-based analysis

**Probabilistic reversal learning task.**   As an overview, we found that paranoia was only associated with decision temperature ($\tau$) and absolute trial-wise prediction errors after adjusting for confounders.

We tested how well several models captured choice behaviour across all participants. These models were variants of the Q-learning model [22–23] with a Softmax response function, so that all models included a decision temperature (higher values mean noisier choice behaviour), and a learning rate ($\lambda$), although some included additional parameters (see Methods). We found that a modified Pearce-Hall model including a 'reset-at-reversal' parameter ($\eta_{pr}$) best accounted for the data while retaining rich enough a parametrization to allow straightforward comparisons across individuals (see methods for full model comparison statistics, equations, and model fitting procedure; S1 Table). We were able to recover all model parameters very well and generate simulated data that closely matched the real data observed (S4 Fig).

Prior to applying statistical controls (model P7a), we found that paranoia was associated with a reduced learning rate (-0.09, 95%CI: -0.16, -0.01) and increased decision temperature (95%CI: 0.17, 95%CI: 0.09, 0.24).

After controlling for general cognitive ability, age, and sex, we found that only decision temperature was associated with paranoia, with all other parameters sharing non-significant relationships (see Table 1; model P7b). As decision temperature can be conflated with model fit, we additionally regressed paranoia against decision temperature, statistical controls, and

**Table 1. Top Model Average of Parameters Associated with Pre-Existing Paranoia in the Probabilistic Reversal Task.** All regression estimates are extracted from Model P6 in the analysis code.

| Parameter | Estimate | Std. Error | 95% CI | | Relative Importance |
|---|---|---|---|---|---|
| | | | lower | Upper | |
| (Intercept) | -0.07 | 0.05 | -0.16 | 0.02 | |
| **Sex** (Male \| Female) | **0.21** | **0.08** | **0.05** | **0.36** | **1** |
| **Decision Temperature ($\tau$)** | **0.13** | **0.04** | **0.05** | **0.20** | **1** |
| Reset-at-reversal ($\eta_{pr}$) | 0.01 | 0.04 | -0.04 | 0.10 | 0.20 |
| Salience ($S$) | 0.01 | 0.02 | -0.04 | 0.01 | 0.22 |
| Memory decay ($\varphi$) | 0.00 | 0.04 | -0.04 | 0.10 | 0.18 |
| Learning Rate ($\lambda$) | -0.02 | 0.03 | -0.13 | 0.03 | 0.46 |
| **Control Questions** | **-0.10** | **0.03** | **-0.17** | **-0.03** | **1** |
| **ICAR Score** | **-0.11** | **0.04** | **-0.18** | **-0.04** | **1** |
| **Age** | **-0.17** | **0.04** | **-0.24** | **-0.10** | **1** |

included the sum loglikelihood score for each participant as an extra regressor (model P8). Decision temperature was still associated with paranoia in this adjusted model (0.11, 95%CI: 0.04, 0.19).

Paranoia was not associated with larger average absolute trial-wise prediction errors (i.e., prediction error size regardless of whether it was positive or negative; 0.10, 95%CI: -0.002, 0.19; model P6). There was an interaction of paranoia with trials pre- and post-reversal, with smaller absolute prediction errors after the reversal in those with higher paranoia compared to before the reversal (-0.25, 95%CI: -0.37, -0.12; model P6).

**Modified repeated reversal dictator game.** To outline, data was best explained by a Bayesian-Belief model that hypothesised that participants' separately weight changes to harmful intent and self-interest attributions following changes to a partner's behaviour. After adjusting for confounders, paranoia was associated with greater uncertainties over a partner's policy ($u\pi$) and stronger priors over harmful intent ($pHI_0$; but not self-interest, $pSI_0$). We found that paranoia was not associated with general, non-specific fixity in attributions ($\eta_{dg}$), but rather was associated with a higher sensitivity to explain changes in behaviour by adjusting SI ($w_{SI}$), but not adjustments to HI ($w_{HI}$).

After comparing original belief-based [18], extended belief-based (Fig 2), and associative social attribution models (see methods and S1 Text), we found the extended belief-based social attribution model best fitted the data—this model allowed participants to weight their explanations of behavioural change through independent adjustments of HI and SI, rather than prior iterations that fixed these parameters. We were able to recapitulate observed data with our winning model (see S7 Fig) and recovered our parameters very well (S11 Fig).

We also replicate prior results [18]: using bootstrapped network analysis we observed positive associations between the strength ($pHI_0$) and uncertainty ($uHI_0$) of the prior over a partner's harmful intent (0.19, 95%CI: 0.11, 0.26), the strength of priors over harmful intent and paranoia (0.13, 95%CI: 0.05, 0.20), and paranoia and uncertainty over a partner's policy ($u\pi$; 0.12, 95%CI: 0.04, 0.20), and a negative association between strength ($pSI_0$) and uncertainty ($uSI_0$) of the prior over a partner's self-interest (-0.11, 95%CI: -0.20, -0.03). We also found a positive relationship between uncertainty over a partner's policy and how much participant's reset their beliefs following a reversal ($\eta_{dg}$; 0.09, 95%CI: 0.01, 0.16; See S12A Fig and S3 Table). An unexpected negative relationship between the strength of priors over harmful intent and uncertainty over a partner's policy (-0.13, 95%CI: -0.21, -0.05) may also exist, suggesting that it is normative to have a more consistent map of a partner if priors over harmful intent are larger.

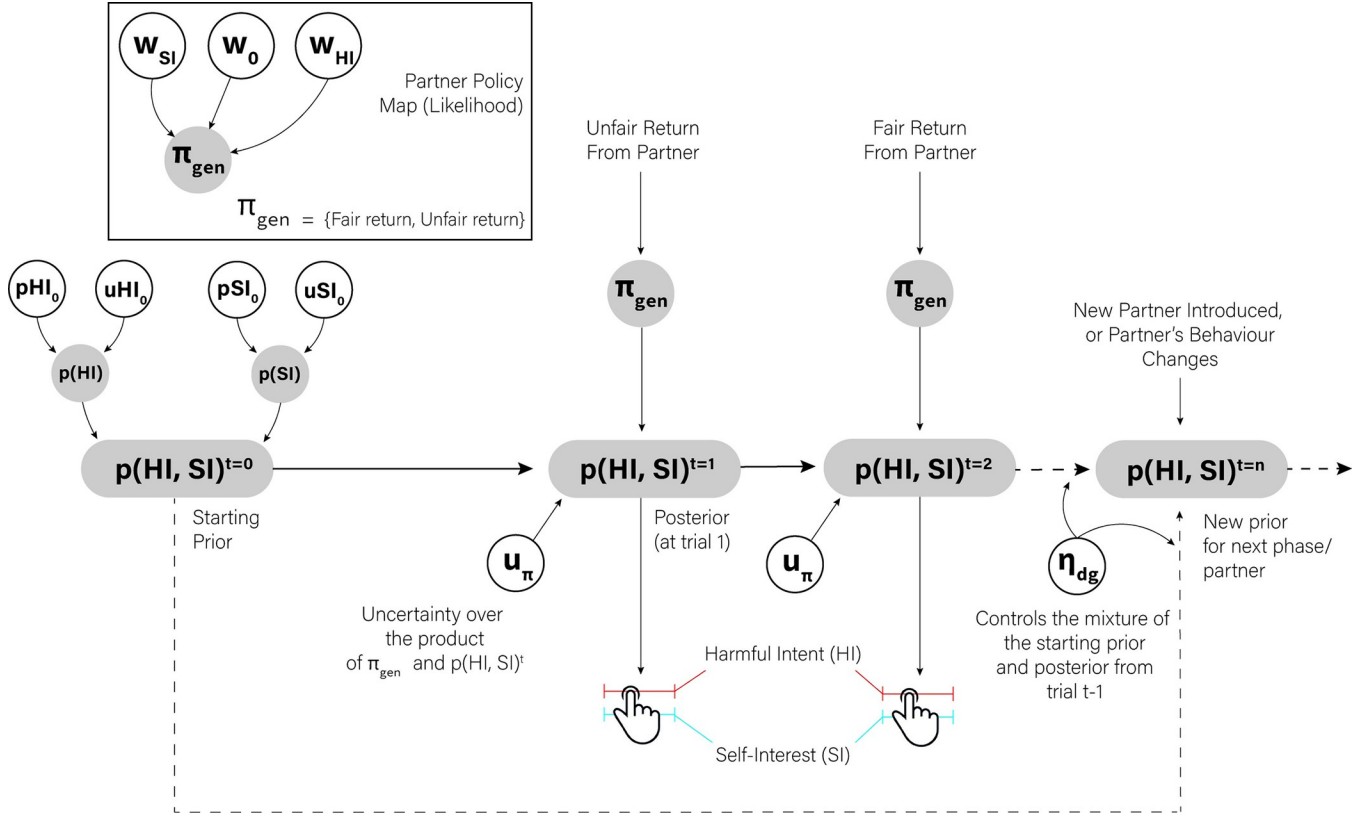

**Fig 2. Extended belief-based social attribution model schematic.** White nodes represent free parameters of the model. Grey shaded nodes represent numerical probability matrices built from free parameters. Thick solid and thick dotted lines represent transitions between trials. Thin solid lines represent the causal influence of a node on another node or variable. The agent or participant updates their initial beliefs (starting prior) about the partner's intentions ($p(HI, SI)^{t=0}$) each trial using their policy matrix of the partner ($\pi_{gen}$) which maps the likelihood between a partner's return to the participant and the partner's true intentions weighted by three free parameters: a policy-map intercept ($w_0$), sensitivity to update self-interest attributions ($w_{SI}$), and sensitivity to update harmful intent attributions ($w_{HI}$). The integration between the likelihood and prior belief from the previous trial is also subject to another free parameter, uncertainty over partner policies ($u\pi$). We assume that upon detecting a change (in this task, a reversal), participants re-set their beliefs, using their priors about people in general (thin dotted line), biased by what they have learnt already about their present partner (reset-at-reversal—$\eta_{dg}$). Both the policy matrix and initial beliefs about the partner are numerical matrices that assigned probabilities to each grid point of values of harmful intent (0–1) and self-interest (0–1). The model can be used to simulate observed attributions of intent given a series of returns, or inverted to infer the parameter values for participants, using experimentally observed attributions.

However, this relationship may be a result of collider bias due to their independent positive relationships with paranoia (S13 Fig) and therefore needs to be interpreted with caution.

Following the generative and replication analysis, we asked how parameters might be associated with paranoia, controlling for age, sex, general cognitive ability, and initial partner behaviour. As expected from our previous study [18] we found that paranoia was associated with higher strength of priors over harmful intent and uncertainty over a partner's policy (Table 2). In contrast to our preregistered predictions, we did not find that the reset-at-reversal parameter was associated with paranoia (which might account for general, non-specific fixity). Instead, we found that paranoia was associated with policy, i.e., the propensity to give unfair returns, being more sensitive to adjustments in self-interest ($w_{SI}$). While this may sound counter intuitive, in fact, greater sensitivity to adjustment self-interest means that those who are more paranoid are more likely to explain changes in behaviour through SI, rather than changing beliefs their beliefs about HI (see S11 Fig for a simulation and illustration of this change with a range of $w_{SI}$ values).

**Table 2. Top Model Average of Parameters Associated with Pre-Existing Paranoia in the Modified Repeated Reversal Dictator Game.** All regression estimates are extracted from Model S5 in the analysis code. NA indicates that the parameter was not included in the final top model.

| Parameter | Estimate | Std. Error | 95% CI | | Relative Importance |
|---|---|---|---|---|---|
| | | | lower | Upper | |
| (Intercept) | -0.06 | 0.05 | -0.15 | 0.03 | |
| **Sex** (Male \| Female) | **0.18** | **0.08** | **0.03** | **0.33** | **1** |
| **Strength of priors over harmful intent ($pHI_0$)** | **0.16** | **0.04** | **0.09** | **0.24** | **1** |
| **Uncertainty over partner policies ($u\pi$)** | **0.17** | **0.04** | **0.10** | **0.25** | **1** |
| **Sensitivity to update self-interest attributions ($w_{SI}$)** | **0.15** | **0.04** | **0.08** | **0.22** | **1** |
| Uncertainty of priors over harmful intent ($uHI_0$) | 0.00 | 0.02 | -0.04 | 0.10 | 0.11 |
| Control Questions | 0.00 | 0.02 | -0.11 | 0.04 | 0.10 |
| Strength of priors over self-interest ($pSI_0$) | 0.00 | 0.01 | -0.09 | 0.06 | 0.08 |
| Reset-at-reversal ($\eta_{dg}$) | 0.00 | 0.02 | -0.10 | 0.04 | 0.11 |
| Uncertainty of priors over self-interest ($uSI_0$) | 0.00 | 0.02 | -0.11 | 0.04 | 0.06 |
| Initial Partner Behaviour (Fair \| Unfair) | 0.00 | 0.02 | -0.12 | 0.17 | 0.08 |
| Sensitivity to update harmful-intent attributions ($w_{HI}$) | NA | NA | NA | NA | NA |
| Policy-map intercept ($w_0$) | -0.06 | 0.04 | -0.14 | 0.01 | 0.89 |
| **ICAR** | **-0.07** | **0.04** | **-0.15** | **-0.00** | **1** |
| **Age** | **-0.16** | **0.04** | **-0.23** | **-0.08** | **1** |

**Association between social and non-social parameters.** Finally, we examined the relationship between derived parameters that shared independent relationships with paranoia across both tasks. In brief, we found that decision temperature ($\tau$) was positively associated with HI (but not SI), the strength of priors over harmful intent of the partner ($pHI_0$; but not $pSI_0$), and pre-existing paranoia.

We initially tested the relationship between decision temperature from the probabilistic reversal learning task and observed attributions in the modified repeated reversal Dictator game. In unadjusted analysis, we found that decision temperature was positively associated with HI (0.14, 95%CI: 0.08, 0.19; model J1a), and negatively associated with SI (-0.07, 95%CI: -0.13, -0.01; model J1a; see Fig 3 for spearman correlations). Adjusting for statistical controls did not influence the effect of HI (0.08, 95%CI: 0.02, 0.13; model J1b) but attenuated the effect of SI (-0.02, -0.09, 0.02; model J1b).

We then tested the associations of all social parameters with decision temperature. Independent spearman correlations suggested that decision temperature was associated with greater strength of priors over the harmful intent ($\rho = 0.16$, $p_{permuted} \sim 0$), uncertainty over partner policies ($\rho = 0.09$, $p_{permuted} = 0.015$), and paranoia ($\rho = 0.16$, $p_{permuted} \sim 0$; See Fig 3). We then regressed all social parameters together against decision temperature. In this model (model J2a), decision temperature was only associated with the strength of priors over harmful intent (0.17, 95%CI: 0.09, 0.24). After including statistical controls (model J2b), decision temperature was still associated with the strength of priors over harmful intent (0.10, 95%CI: 0.02, 0.18). After introducing paranoia (model J2c), decision temperature was associated with both paranoia (0.11, 95%CI: 0.03, 0.18) and the strength of priors over harmful intent (0.09, 95%CI: 0.01, 0.16; see S4 Table for all estimates and 95%CIs).

## Discussion

We assessed the association between social and non-social reversal learning, and the impact of paranoia on both, in a large sample of non-clinical individuals. In the non-social task, paranoia was associated with suboptimal choices following a reversal, and greater decision temperature. In the social task, attributional model comparison uncovered that a Bayesian-Belief model that

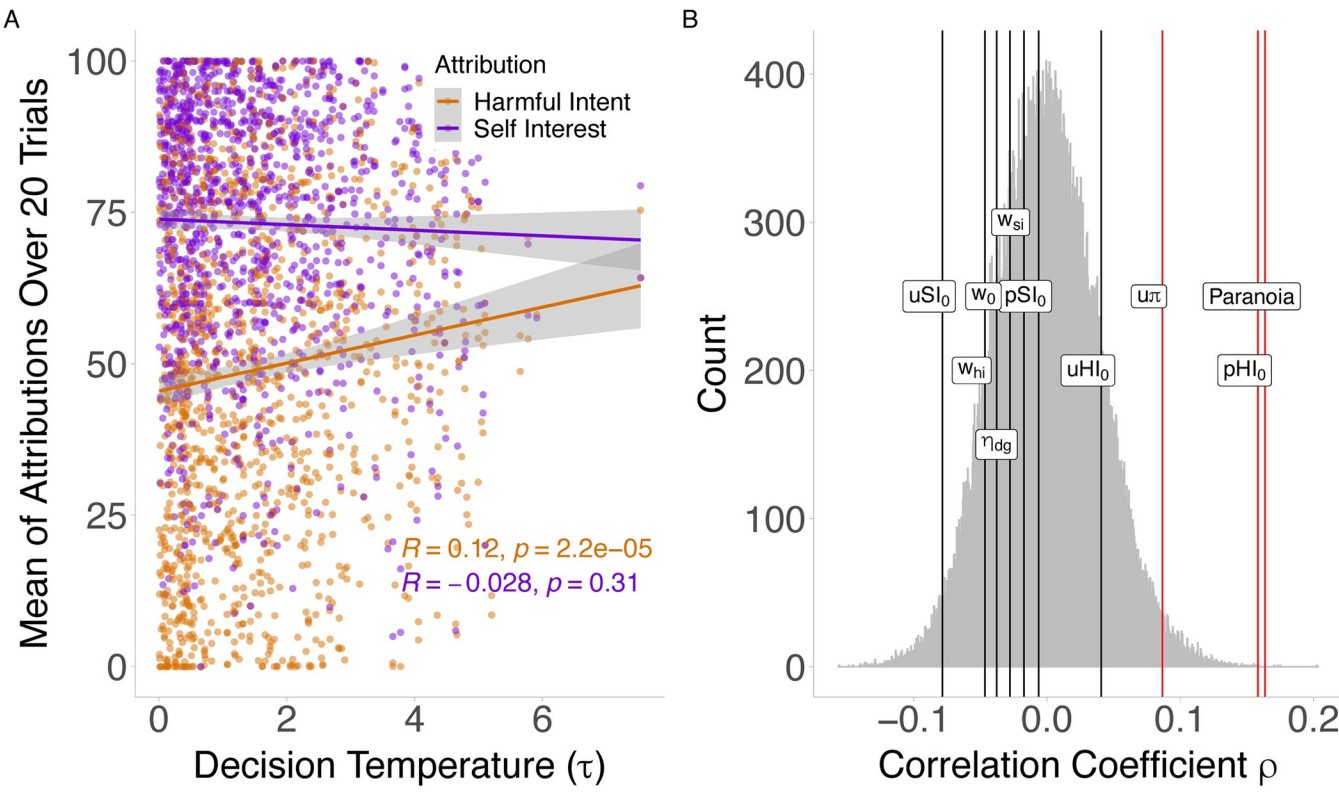

**Fig 3. The relationship between decision temperature, attributions, and social task parameters.** (A) Spearman correlations between decision temperature and mean attributions observed summed across 20 trials for each participant. (B) Permutation analysis of the relationship between decision temperature, and computational model-based parameters from the winning model and pre-existing paranoia. The grey distribution represents the null distribution following random sampling of the population for each Spearman pairwise correlation. The true Spearman correlations of each social parameters against tau are depicted for each parameter. Only the strength of prior beliefs over harmful intent ($pHI_0$; $\rho = 0.16$, $p_{permuted} \sim 0$), uncertainty over partner policies ($u\pi$; $\rho = 0.09$, $p_{permuted} = 0.015$), and paranoia ($\rho = 0.16$, $p_{permuted} \sim 0$) were associated with decision temperature. Red lines denote that the observed correlation with tau is very unlikely due to chance ($p < 0.05$). Black lines denote the observed correlation is more likely due to chance ($p > 0.05$).

used separate weights on harmful intent and self-interest attributions to explain a partner's behavioural change best fit the data. From this we found that paranoia was associated with policy uncertainty, larger strength of priors over beliefs about a partner's harmful intent (but not self-interest), and that paranoia was associated with greater sensitivity to explain a partner's behavioural change through self-interest rather than harmful intent. Finally, we observed that decision temperature in the non-social task was associated with larger strength of priors over a partner's harmful intent (but not self-interest), harmful intent attributions over all trials, and uncertainty over partner policies in the social task, and with pre-existing paranoid beliefs. Our model and data raise hypotheses that may bridge general reinforcement learning and specific phenomenological explanations of the paranoia and allow experimental testing of predictions with formalised computational targets.

In line with predictions, we found elevated decision temperature in the non-social task in those with higher paranoia, although the interpretation of this is not straight forward. Higher decision temperature can be indicative of different causes: it could be signs of information-seeking behaviours (e.g., strategic or directed), or instead random stochastic exploration without any reward or information gain [24–25]. The former would reflect lower-valued options being selected less frequently over time, and the latter demonstrated by frequent switching trial to trial with repetitions of the same actions regardless of reward. Prior work has found

noisier decision making is associated with high risk and clinical participants after initial reversals [1–2], in those reporting psychotic experiences [26], and in healthy populations with higher paranoia [3,24]—these latter studies in particularly found larger win-switch rates across all trials in addition to larger decision noise. This would suggest decision temperature in paranoia might be related to more random behaviour. However, in one study, global impairment was found to confound random trial by trial switching behaviour: those with a schizophrenia diagnosis but higher in verbal and working memory showed win-stay behaviour no different to healthy controls [3]. Converging with this finding, and using a larger sample than previously employed, we found no increased win-switch or lose-stay rates when examined across all trials after statistical adjustment for fluid intelligence. Instead, we found increased win-switch rates and choosing suboptimal choices in the more paranoid only after reversals. Along with prior work, we suggest: 1) paranoia is related to directed exploratory behaviour when the environment changes with the overestimation of previously optimal cards and 2) optimal choices are not ignored in those who are more paranoid but may instead take longer on average to become exploited, leaving more room for ambiguity.

We replicated key parameter relationships from the social model [18]. We found that larger priors over beliefs about a partner's harmful intent conferred greater prior uncertainty over harmful intent, whereas the opposite was true for self-interest: larger prior beliefs concerning a partner's self-interest were held with more certainty. We also replicated the relationship between paranoia and uncertainty regarding how strongly a partner's actions relate to their true intentions. Unexpectedly, we found that uncertainty over partner policies were positively, rather than negatively, associated with the switch parameter. This means that as individuals become more uncertain over partner behaviour, they become more rigid in their attributional changes after the reversal. This disparity may have been due to our different task design and our extended model: the original task was used to explain between-partner adaptation [18] whereas in this task we model within-partner adaptation. Therefore, we are estimating qualitatively different changes in behaviour. This suggests that believing the same partner to be inconsistent with their actions is linked to less inferential flexibility when a partner's behaviour changes.

Unexpectedly we found that paranoia was associated with a greater weight being placed on a partner's policy of self-interest, rather than a general fixity in attributional dynamics. Our winning model allowed participants to hold asymmetric sensitivities to whether fluctuations in a partner's behaviour was attributed to changes in their underlying harmful intent or self-interest. This won over and above our previous model [18] which held the partner's policy map with fixed parameters. Contrary to our prior hypothesis, rigidity over harmful intent was not due to a lack of sensitivity to changing partner behaviour, but rather a hypersensitivity to explain changes in behaviour with counter factual reasoning. Specifically, simulations using a range of $w_{SI}$ values demonstrated that this led to greater flexibility over self-interest attributions but not harmful intent attributions following a change in behaviour from a partner. Our results are congenial with models of general belief fixity (cf. [27]) that explain delusional maintenance through a desire to dismiss incongruent, counterfactual evidence with alternative hypotheses, although our model allows for the measurement of clinically relevant phenomena.

Decision temperature in a non-social task was associated with larger priors over harmful intent, uncertainty over beliefs about a partner in unadjusted analyses, and pre-existing paranoia, but not parameters that control self-interest attributions. Given the empirical relationship between pre-existing paranoid beliefs and psychosis on uncertainty over environments [2, 3, 7, 21, 28–30] it is unsurprising that both non-social and social uncertainties are jointly related to paranoia in this present experiment, although we demonstrate this explicitly in

relation to pre-existing paranoia and attributions in the moment. There may be several reasons for these associations.

First, there may be a common biological mechanism responsible for the expression of uncertainty in both non-social and social contexts. Prior theoretical work explains the relationship between dopamine (dys)regulation, psychosis, and probabilistic reasoning [11,13], and empirical evidence has supported the common role of dopamine (dys)regulation in influencing uncertainty about the world [3, 31], the learning of information from primary vs secondary sources [32], adjusting harmful intent and externalising attributions [33–34], and increasing psychotic experiences [35–36]. While we do not use psychopharmacological manipulations in this paper, evidence to date is consistent with dopaminergic signalling being causally implicated in the basic computational processes underlying decision making (e.g., decision temperature) and should also be tested to assess whether changes to dopamine signalling also underlies uncertainty about a social partner, and whether this added uncertainty mediates increases in harmful intent attributions.

A second, non-mutually exclusive explanation may be that increases in non-social decision temperature is a response to second-order social uncertainty made about the experimenters. In one study, paranoia was found to increase belief that a cards task was intentionally sabotaging the participant [21] and may have been responsible for the studies reported increase in overall win-switch behaviour. This raises the question: to what extent can 'non-social' task designs can be considered to measure non-social behaviour uncorrupted by agentive attributions? Not only is this question important for psychological measurement of behaviour, but the attribution of agency also has implications when associating neural activity with performance in tasks: prior work has demonstrated differential temporal-parietal junction activity as part of the 'mentalising network' dependent on whether a participant is perceiving to play against a computer, robot, or human social partner [37]. A way to remedy this would be to control for first- and second-order agency attributions, i.e., whether a partner was perceived to be 'real', or the inference that experimenters were intentionally trying to mislead the participant, respectively.

Our belief-based model explicitly defines parameters that capture sociocognitive processes outlined in prior descriptive theory that explain the formation and maintenance of persecutory ideation. Rich state space models are required to capture the added complexity of a social interaction over and above those which quantify leaner learning processes [17, 38] belief-based model contributes to this theoretical requirement. First, uncertainty over others or over the self as a prerequisite for persecutory ideation has been theoretically [13–16] and empirically [7, 39–40] supported. Our model identifies the consistency to which we hold our internal statistical map of social others ($u\pi$), which when elevated, causes greater uncertainty in a participant's beliefs about a partner. Secondly, persecutory ideation has been robustly associated with externalised attributions of harmful intent [15, 34, 41–42]. The degree to which one holds strong beliefs of harmful intent at the start of an interaction is formalised in our model ($pHI_0$), which when increased, leads to higher initial expectations of harmful intent from a partner before interaction. Importantly, this parameter can be dissociated from priors over other, qualitatively different attributions ($pSI_0$). Finally, cognitive models of persecutory delusions [16] and *in silico* demonstrations [27, 43] suggest disconfirmatory evidence is explained away with alternatives when evidence deviates from a delusional belief. In our model, two parameters ($w_{HI}$, $w_{SI}$) quantify attributional flexibility which may be used to probe how pre-existing beliefs bias asymmetric interpretations of behavioural change.

We offer several predictions: 1) as demonstrated in our non-social task, it may be that healthy participants with higher paranoia need longer to gauge a social partner's intentions, but over longer periods may eventually reach the same conclusions as the group. We predict

that when partners become more consistent in their social behaviours, a high-paranoia participant's map of an interaction partner will become more precise ($u\pi$ will reduce). 2) In line with prior work examining the influence of cannabis on paranoia [44] and the specific role of dopamine modulation on attributions of harmful intent [45], we predict dopamine potentiation will increase uncertainty over partner policies ($u\pi$) and the strength of priors over harmful intent ($pHI_0$), but not the strength of priors over self-interest ($pSI_0$). 3) On a neural there is evidence that social context may be biologically realised through the engagement of different structures [46], including the dorsomedial prefrontal cortex where social computations may be implemented [9]. We predict that dopaminergic changes that underlying learning in multiple contexts may lead to context specific effects (e.g., social vs non-social learning) such as a participant's uncertainty over their partner ($u\pi$). 4) In clinical populations with a history of aversive or traumatic social environments during childhood and adolescence, belief maps will be more uncertain ($u\pi$ will remain high), harmful intent attributions will remain higher (higher initial priors, $pHI_0$) and less flexible (lower $w_{HI}$ or higher $w_{SI}$) than that of healthy controls.

We note three limitations. While the similarity of constructs across different, ecologically valid tasks is a strength of our study, it also means we cannot directly compare behaviour in one task to another as they require different models/task content. An alternative would be to create a 'social' version of a non-social task (e.g., [21]). Suthaharan and colleagues [21] aimed to assess whether probabilistic reversal learning in those with higher paranoia differed between card decks that were and were not putatively controlled by a social agent, finding no difference in parameter estimates in those more paranoid across both tasks. However, tasks such as that used by Suthaharan and colleagues may be measuring social observation more than they are measuring social interaction; the latter requires an interaction partner's behaviour to be 'online' (i.e., the decisions of the partner result in outcomes for both the partner and the participant; [47]). Secondly, we use a non-clinical population, and it is unclear whether the parameter estimates derived from our models in those with higher pre-existing paranoia would exist in clinical populations, although as mentioned above, we make some predictions about how the transition to clinical populations may unfold. Finally, we did not use varying volatility in our non-social task, keeping the same probabilistic environment with a single reversal. It may be that our single reversal meant participants had less time to build up expectations of contingency changes, despite not being told when the reversal might occur.

## Methods

### Ethics statement

The experiments were internally reviewed and approved by the Research Ethics Committee at King's College London, UK (ref: RESCM-19/20-0603). Participants gave consent by ticking checkboxes online following the information sheet, and prior to the administration of questionnaires or tasks.

### Participants

As with prior experiments (e.g., [34, 48]), demographics (age, sex, education), pre-existing paranoia (using the persecutory subscale of the R-GPTS-B; [19]) and general cognitive ability measured using ICAR matrices ([20]) was measured seven days prior to the experimental paradigms.

We recruited 750 participants at baseline. We lost 54 participants in the follow up between baseline questionnaires and administration of the tasks. 7 participants had incomplete data for at least one of the tasks. Therefore, we analysed 693 participants (66% female) for the modified

repeated reversal Dictator game, 692 participants for the probabilistic reversal learning task (66% female), and 689 for the joint analysis. Data were collected in September 2020 through Prolific Academic. All participants were aged between 18–65, had no prior or current psychiatric or neurological diagnosis (established through screening tools on Prolific academic during population filtering), were fluent in English, and were residents of the UK.

## Paradigms

Participants took part in two tasks during the experimental phase. These were the probabilistic reversal learning tasks and modified repeated reversal Dictator Game.

The probabilistic reversal learning tasks presented three symbols to the participants over 60 trials. Symbols could either provide +10 or -5 points. They were instructed at the start that there would be one symbol that had a high chance (80%), one had an even chance (50%), and one a low chance (20%) of providing +10 points. Participants were also told that the symbol contingencies could change at any point during the game. Halfway through the game (after trial 30), participants were asked to explicitly choose which symbol they thought provided the highest probability of giving points. After trial 30, the contingencies of the card changed for the last 30 trials, such that the lowest probability card became the high probability card, the highest probability card became the even probability card, and the even probability card became the low probability card. At the end, participants were once again asked which symbol they thought had been providing the most points.

The modified repeated reversal Dictator game comprised 20 trials. In the task, each participant was paired with a partner, with the partner represented by different avatars to than the participant. The 'social' game was based on a modified Dictator Game [49]. In this game, the participant's partner was given 10 points in each trial and could choose whether to split this equally with the participant or to keep the points for themselves.

After each human decision, participants rated on a scale of 0–100, initialised at 50, how much they believed their partner's intentions were to reduce their bonus, and rated on (a separate scale of 0–100, initialised at 50) how much they believed their partner's intentions were to try and earn as much money as possible for themselves (hereafter 'self-interest').

Participant would either be matched with initially unfair humans (80/20 probability of not splitting the points) or initially fair humans (80/20 probability of splitting the points). After trial 10 if their partner had been unfair their policy would change to being fair (with a probability of 80/20 fair returns), and vice versa.

After taking part in the social task, participants were assigned to the role of the dictator in a final game. These dictator decisions were not used for analysis but were collected for ex-post matching to truthfully inform participants that their partner's decisions in the social game were real (c.f. [50]).

## Preregistered hypotheses

Probabilistic reversal learning task (https://aspredicted.org/57p5e.pdf) and modified repeated reversal Dictator game (https://aspredicted.org/ds9bf.pdf) predictions were registered online at AsPredicted.org.

We deviate from our preregistered predictions by using general linear models rather than cumulative link models for attributional analysis and deviate through the insertion of interactions stepwise–we felt this to be more interpretable than assessing all interactions at once. In the social task, we included unplanned analyses not recorded in preregistered predictions to better explore the relationship of paranoia to social task parameters, and to explore the interrelationship between non-social and social task parameters.

## Behavioural analysis

All statistics reported in the text are standardised regression coefficients following linear model averaging (to control for variable order and to find the most parsimonious, adjusted regression model) and reported with their 95% confidence intervals, as per (*b*, 95%CI: lower bound, upper bound). All model code in the text is included in the analysis code posted online for cross checking and replication.

All linear mixed models were constructed using the 'LME4' package (v1.1–23) and averaged using the 'MuMln' package (v1.43.17) with data wrangling using 'tidyR' (v1.1.2) and plotting using 'ggplot2' (v3.3.3) in R (Version 4.0.0, 2020/04/24) on a mac OS (Big Sur v11.1). All continuous variables were centred and scaled.

For unadjusted analyses, when outcomes were binary, we used general linear mixed models, and when outcomes were continuous, we used linear mixed models, both with ID used as a random variable.

For adjusted analyses we used general linear (when outcomes were coded as binary 1/0 responses) and mixed linear regression models (when outcomes were continuous) for numeric variables of interest. We analysed each model using multi-model selection with model averaging. The Akaike information criterion, corrected for small sample sizes (AICc), was used to evaluate models, with lower AICc values indicating a better fit [51]. The best models are those with the lowest AICc value. To adjust for the intrinsic uncertainty over which model is the true 'best' model, we averaged over the models in the top model set to generate model-averaged effect sizes and confidence intervals [52]. In addition, parameter estimates, and confidence intervals are provided with the full global model to robustly report a variable's effect in a model [53].

Win-switch and lose-stay behaviour was calculated as in a previous study [21]. Win-switch rates were calculated as the number of times a participant switched options after receiving positive feedback, divided by the total number of trials where they received positive feedback. Lose-stay rates were calculated as the number of times participants stayed on an option after receiving negative feedback, divided by the total number of times they received negative feedback.

Importantly, we planned to control for general cognitive ability and task comprehension in our modelling. General cognitive ability has been previously identified as a confounder of the association between probabilistic reasoning using a canonical beads task and paranoia [54]. Likewise, not assessing whether participants recruited in online samples are attentive or understand the task can lead to spurious correlations [55]. To control for both the possibility that results may arise from 1) poorer general cognitive ability or 2) poor task comprehension instead of pre-existing paranoia we include a measure of non-verbal cognitive ability (ICAR matrices; [20]).

## Computational modelling

**Probabilistic reversal learning task.**   As participants were aware that the task was divided in two blocks, they were more likely to suspect that a change could have taken place between blocks, despite instructions stating that reversals may occur at any moment. Inspired by non-associative change-detection models [6], we tested whether a reset parameter ($\eta_{pr}$) by which participants reset the values of the cards towards the mean value at the point of reversal (trial 30) improved model-fit, over, and above mechanisms used to adapt learning rates in previously successful associative models of reversal learning [56]. The reset parameter thus captured descriptively (rather than through a detailed change-point detection algorithm) the

extent to which participants specifically responded to the reversal. At the same time, we tested whether learning rates were adjusted through a Pearce-Hall salience mechanism [56].

We also considered a potential memory parameter ($\varphi$) that could account for the decay in unobserved symbol values, a lapse rate parameter ($\zeta$), or a separate learning rate ($\lambda_2$) that allowed the learning rate to change from block 1 (before reversal) to block 2 (after reversal). We thus compared models with 2 to 7 parameters.

In addition to our range of Q learning variations, we considered pure 'win-stay, lose-switch' models and Pearce-Hall models as nested within our complex RW model (setting $\tau = 0.01$, $\lambda = 0.99$ for WSLS) and keeping parameters $\theta = [\tau, \lambda, S]$ for Pearce-Hall. We first used grid-fit and simulated annealing procedures to increase the chance of fitting to the global optima in maximum-likelihood estimations for each model for every participant, and then refined parameter estimates by gradient descent using MAP estimation procedures with weak regularising priors.

**Formalism.** We constructed a variation on the classic Q-learning model system (Watkins & Dayan, 1992) that computes the subjective internal value of a series of agents or symbols in the environment. The classic model computes a value function for each option $Q_c$, in our case for three symbols. $Q_c^{t=0}$ was initialised to 2.5 (the mean reward expected given that each symbol has $P$ probability of giving a +10 or -5 point outcome). Then on every action taken, after a participant has chosen option $c$ on trial $t$ and received an outcome $r$, the value of each $Q_{\hat{c}}^t$ is updated as follows:

$$Q_{\hat{c}}^t = Q_{\hat{c}}^{t-1} + \lambda * (r^t - Q_{\hat{c}}^{t-1}) \tag{1}$$

$\lambda$ is the learning rate over the entire task which was calculated using the single parameter $\lambda_1$ in models that used a single learning rate for all 60 trials. We also fitted models where the learning rate was determined by a new free parameter, $\lambda_2$, after trial 31.

For the Pearce-Hall modification [57] of the learning rate, we adjusted the learning rate in Eq1. by a salience parameter, where *Salience* for trial $t$ given action $Q_{\hat{c}}^t$ is defined by:

$$Salience^t = S * |PE| + [(1 - S) * Salience^{t-1}]$$

$$Learning\ rate = Salience^t * \lambda \tag{2}$$

$$Q_{\hat{c}}^t = Q_{\hat{c}}^{t-1} + Learning\ rate * PE$$

This replaces Eq 1. Where $PE = (r^t - Q_c^{t-1})$ for the previous trial, as per Eq 1. To implement our memory parameter, $\varphi$, we decayed all $Q_c^t$ values that were not selected (-c) for any given trial $t$, towards the mean value (2.5) of possible returns. This replaced Eq 1. Where $\in \{c_1, c_2, c_3\}$:

$$Q_c^t = \begin{cases} Q_{\hat{c}}^{t-1} + \lambda * (r^t - Q_{\hat{c}}^{t-1}) \ if\ \hat{c} = chosen \\ 2.5 - \varphi * (2.5 - Q_{\hat{c}}^{t-1}) \ if\ \hat{c} \neq chosen \end{cases} \tag{3}$$

To implement our reset parameter, $\eta_{pr}$, we shifted all Q values towards the same mean value, 2.5, by $\eta_{pr}$ before trial 31 (immediately after the reversal):

$$\bar{Q}^{t=30} = Q^{t=30} + [\eta_{pr} * (2.5 - Q^{t=30})] \tag{4}$$

$\bar{Q}^{t=30}$ then became the new prior for trial 31. Policy probabilities for any given trial were calculated using a SoftMax function of the current Q value at trial $t$ subject to a decision

temperature, $\tau$:

$$p(\hat{c}^t = c) = \frac{\exp\left(\frac{Q_c^{t-1}}{\tau}\right)}{\sum_{c' \in \{C_1, C_2, C_3\}} \exp\left(\frac{Q_{c'}^{t-1}}{\tau}\right)} \tag{5}$$

Finally, we also allowed for a lapse parameter, $\zeta$. This allows for processes that are independent of motivated choice, as estimated by Eq 5, so that in a fraction $\zeta$ of trials an unknown process, approximated by a flat distribution over the choices, is assumed to operate (for example, a complete lapse of attention):

$$\pi^t = \frac{\zeta}{3} + (1 - \zeta) * p(\hat{c}^t = c)$$

$$LL = \log\left(\pi^t\right) \tag{6}$$

**Modified repeated reversal dictator game.** The original model formalism used in the analysis of the social task can be found in a previous paper [18].

We compared a previously derived probabilistic Bayesian model, augmented by a 'switching parameter' ($\eta_{dg}$) analogous to the resetting parameter above, to fit to the modified repeated reversal Dictator Game [18]. We also compared several associative models inspired by prior work modelling self-esteem [58]; this set of associative models employ the same conceptual structure as non-social associative learning models (see S1 Text for the full formalism of all social associative models). In essence, this suite of models used logistic mappings, each including intercept ($wHI_0$, $wSI_0$) and weighting ($w_{HI}$, $w_{SI}$) parameters to predict each attribution with a single 'expected social value' as independent variable–a cached, Markovian latent variable. This value was subject to an initial expected social value parameter ($ESV_0$) and was updated through a learning rate ($\alpha$). An attribution noise parameter ($\sigma$) completed the generative model. We also considered two-$\eta$ models, where detecting a change (reversal) had a different impact depending on harm- vs. self-interest intent. Finally, we built a set of models using a similar, logistic mapping between the partner's attributes and their policy (the likelihood function) based on the belief-based (Bayesian) models of our previous work [18]. This was possible as the more powerful manipulation of contingency reversal allowed for individual fitting of parameters of the attribute-policy map for each person ($w_0$, $w_{HI}$, $w_{SI}$; see Fig 2).

The models were initially fitted with Maximum A Posteriori (MAP) estimation on 100 random participants, i.e. penalizing maximum likelihood with a weak, regularizing prior restricting parameter values to their psychologically meaningful ranges (e.g. learning rate between 0 and 1, etc.). A simulated annealing approach on parameter values was followed by gradient-ascent on MAP to minimize the chance of missing important MAP maxima. A belief-based model with a single switching parameter ($\eta_{dg}$) best fitted the data (S4 Table) when assessing the BIC and AIC values from the discovery subset (n = 100) of participants.

We then sought to fit all participants. As all belief-based models showed better fits than associative models, we applied concurrent Bayesian model comparison [59] to no-, one- and two- $\eta_{dg}$ belief-based models, in addition to the best fitting associative model, to look for participants better accounted by an associative framework (see methods). We fitted each series of models on four groups within our population, divided by high/low paranoia and high/low general cognitive ability. This was to ensure group-level empirical priors were able to capture the potential nuance within each class of participant.

We observed that the belief-based model with a single switching parameter still fitted the data best (S8 Fig). We assessed the candidate winning model for predictive and generative performance. The ability of a model to simulate data is necessary to assess its validity and falsification [60–61]. This centred around our ability to replicate our effects documented from our reported behavioural results in this same paper. We then aimed to assess our model fitting by using the log-likelihood values across trials, dictators, and divisions of GPTS score (z scaled, continuous GPTS scores). Following this, we aimed to statistically interrogate the generated data in the same manner as we did with the behavioural data.

**Winning model formalism.** We model effective beliefs about dictator's attributes as ranging along two dimensions, harmful intent, and self-interest attributions. We can discretise them into Likert-like bins (Nb = 9). Here, we discretised along 9 bins, from 'totally altruistic' ($HI = 1$, $SI = 1$) to 'totally antisocial' ($HI = 9$, $SI = 9$). The prior beliefs about Others formed the most important part of our modelling, parametrized by a central tendency parameter $pHI_0$, $pSI_0$ and an uncertainty $uHI_0$, $uSI_0$ along each dimension. Inference over such discrete distributions can be conveniently parametrized the Binomial distribution with $n$ bins and parameter $p$, sharpened (or blunted) by an uncertainty parameter $u$:

$$Pk \propto Bin(k; p, n)^u := NB(k; p, u, n) \qquad (7)$$

When the exponent in Eq 7 is greater than 1, the distribution keeps the same mode but is sharpened; when less than 1, it is blunted. The prior belief over both $HI$ and $SI$ can then be written as a product of the independent prior probabilities, $p(HI)^{t=0} * p(SI)^{t=0}$. This assumption of independence is conservative, minimizing the number of free parameters:

$$p(HI)^{t=0} = Bin(HI; pHI_0, uHI_0, Nb)$$

$$p(SI)^{t=0} = Bin(SI; pSI_0, uSI_0, Nb)$$

$$p(HI, SI)^{t=0} = p(HI)^{t=0} p(SI)^{t=0}$$

To make inferences based on the feedback they get from dictators, participants must also hold a correspondence between attributes and behaviours. We emphasise that participants hold maps *from attributes to behaviour*, and not directly from observations of returns to attributes. Therefore, participants must invert these maps to update their beliefs, which will typically result in asymmetric belief updates depending on further detail (so that Eq 7 uses full joint probabilities, breaking the initial independence). To build a map from attributes to behaviour that could capture a full range of possibilities we first provided for a range of possible dictator behaviours, discretising returns using a similar resolution as attitudes. We implemented this general template map $\pi_{gen}$ using free parameters, where $\pi_{gen}$ is a Nb x Nb numerical matrix. The corresponding equations (Eq 8) are given below for completeness:

$$\pi_{gen}(r = 0; HI, SI) = \sigma\left(w_0 + \left[w_{HI} * \left(HI - \frac{Nb+1}{2}\right)\right] + \left[w_{SI} * \left(SI - \frac{Nb+1}{2}\right)\right]\right)$$

$$\pi_{gen}(r = 0.5; HI, SI) = 1 - \pi_{gen}(r = 0.5; HI, SI) \qquad (8)$$

where σ is a logistic sigmoid.

For each potential attribute pair (*HI*, *SI*) of the Dictator (which is a numerical matrix) we multiply the likelihood, $\pi(r; HI, SI)$, by the prior, $p(HI, SI)^{t-1}$:

$$p(HI, SI)^t = \frac{\pi(r; HI, SI)p(HI, SI)^{t-1}}{\sum_{HI',SI'} \pi(r; HI', SI')p(HI', SI')^{t-1}} \qquad (9)$$

This completes the participants' generative beliefs of the Dictator's behaviour, and provides for exact, numerically tractable Bayesian updates in the beliefs of the participant when they receive feedback. One additional parameter was introduced, to quantify individual variation in the consistency agents expected between beliefs and behaviours. Based on previous work, a small, fixed lapse rate $\xi = 0.02/n^2$ was also added to increase numerical stability. This was another noise or uncertainty parameter $u_\pi$, over the dictator's policies. We thus used:

$$p(\widehat{HI, SI})^t \propto p(HI, SI)^{\frac{1}{u_\pi}} + \xi \qquad (10)$$

Where $p(H\hat{I}, SI)^t$ then becomes the generative belief distribution to emit attributions for each trial. We note that in our experiment it is not possible to clearly distinguish between uncertainty participants display due to their own noisy cognition, as opposed to noisy decision-making that they expect their partners to display. In our case, both would result in greater participant uncertainty and noisier reporting of inferred attributes.

We also considered that participants inform their beliefs about the change in a partner's policy observed after trial 10 by what they learnt about the first set of outcomes. The simplest approximation is to add a small admixture of the posterior beliefs about the initial actions of the Dictator to the priors they used for the new action policy, weighing this posterior by an individually fitted learning rate $\eta_{dg}$. This then creates a new prior $(p(H\bar{I}, SI)^{t=10})$ to be used in Eq 9. This parameter was used to assess perseveration of beliefs between trials 1–10 and 11–20:

$$\overline{p(HI, SI)}^{t=10} = (p(HI, SI)^{t=0} * [1 - \eta_{dg}]) + (p(HI, SI)^{t=10} * \eta_{dg}) \qquad (11)$$

**Network analysis.** To assess the interrelationship of social and non-social parameters, and to replicate prior work, we applied regularised Gaussian Graphical Model estimation techniques implemented in the R programming language through the 'bootnet' and 'qgraph' libraries [62] using the 'huge' nonparanormal function. Nonparanormal network analyses relax the assumption of normally distributed variables when estimated regularised network and were appropriate given several our parameters were non-normally distributed (S3 and S6 Figs; [63–64]). Networks in this sense are the conditional relationships (edges) between variables (nodes). Networks that were estimated using 'bootnet' apply Least-Absolute Shrinkage and Selection Operator that shrinks very small edges to zero.

We generated a network to replicate our prior work [18]. We computed edge-weight accuracy and node stability using bootstrapping with the 'bootnet' function [62]. While somewhat arbitrary, simulation studies suggest that node stability metrics should be no lower than 0.25 and ideally above 0.5; these figures represent the correlation stability coefficient of a network, and the maximum cases that can be dropped to retain a correlation between the original centrality indices and the case-dropped networks on subsets of 0.7 or higher (CS(Cor = 0.7); [62]). The replication network demonstrated adequate stability (CS(Cor = 0.7) = 0.361 for all statistics) and robust bootstrapped edge estimates (S13 Fig).

## Supporting information

**S1 Fig. Behaviour of the participants in the probabilistic reasoning task.** Top panel: relationship of paranoia and ICAR total score with the proportion of correct cards chosen in each block. Bottom panel: Sum of each chosen card by paranoia and ICAR total score for each block. In Block 1, Card 1 was the optimal card to choose with an 80/20 probability of reward. In Block 2, Card 3 was the optimal card to choose, with 80/20 probability of reward. (DOCX)

**S2 Fig. Probability of choosing a particular card in each block for high and low paranoia.** In Block 1, Card 1 was the optimal choice with an 80/20 probability of reward. In Block 2, Card 3 was the optimal choice, with 80/20 probability of reward. This graph demonstrates that those with higher paranoia were significantly and more consistently likely to choose the suboptimal 20/80 card (Card 2) in block two, and significantly less and more consistently likely to ignore the optimal card (Card 3) in block 2. However, those with higher paranoia were still able to learn which was the more optimal card by the end of block 2. * = $p<0.05$, ** = $p<0.01$ *** = $p<0.001$. (DOCX)

**S3 Fig. Histogram and point distributions of the individual-level fitted parameters derived from the computation model (Probabilistic reversal learning model).** (A) Our model was able to recapitulate the real data well. The real (Q1 –Q3) and simulated (simQ1 –simQ3) Q values generated by the model for each trial across all participants for each different symbol. (B) All parameters were recovered very well. Correlation matrix showing the Pearson correlations between the real (X axis) and recovered (Y axis) parameter. (C) The 5-parameter model produced equivalent to better BIC values compares to the 3-parameter core model. In these plots, blue dots below the line indicate better fit than the reference model (model 3) and above the line indicate the reference model fits better. Correlation comparisons between the BIC values for each alternative model (named in the facet title) and the core 3-parameter model (X axis); reference lines on each plot indicate +/- 6 and +/- 10 BIC values. Models 4 and 5 were not significantly different in individual BIC values from Model 3 ($\chi^2(2) = 2.13$, $p = 0.345$). (DOCX)

**S4 Fig. Probabilistic Reversal Learning Model Fit and Recovery.** X = non-significant relationship. (DOCX)

**S5 Fig. Probabilistic Reversal Learning Partial Correlation Matrix.** (A) Sum loglikelihood for each integer of pre-existing paranoia. Grey horizontal line indicates the sum loglikelihood at which the model is predicting the data by chance. (B) Sum loglikelihood for each integer of ICAR score. Grey horizontal line indicates the sum loglikelihood at which the model is predicting the data by chance. (C) Distribution of sum loglikelihood for each social condition. Grey vertical line indicates the sum loglikelihood at which the model is predicting the data by chance. (D) Correlation between real and simulated harmful intent and self-interest attributions. (E) Averaged real (grey) and simulated (coloured) harmful intent and self-interest attribution for each condition across all trials. Analysis of simulated data using a mixed effects model with ID as a random variable suggested pre-existing paranoia was positively associated with harmful intent (0.11, 95%CI, 0.05, 0.16; model S5a) but not self-interest (-0.02, 95%CI: -0.07, 0.03; model S5b), and being paired with an initially unfair Dictator did not influence harmful intent attributions, but led to larger self-interest attributions (0.27, 95%CI, 0.18, 0.37;

model S4b; see S6 Fig for comparison with real data across both conditions).
(DOCX)

**S6 Fig. Smoothed posterior density distributions of the individual-level fitted parameters derived from the hierarchical Bayesian fit (using CBM; modified repeated reversal Dictator Game).**
(DOCX)

**S7 Fig. Social model assessment.** (A) Sum loglikelihood for each integer of pre-existing paranoia. Grey horizontal line indicates the sum loglikelihood at which the model is predicting the data by chance. (B) Sum loglikelihood for each integer of ICAR score. Grey horizontal line indicates the sum loglikelihood at which the model is predicting the data by chance. (C) Distribution of sum loglikelihood for each social condition. Grey vertical line indicates the sum loglikelihood at which the model is predicting the data by chance. (D) Correlation between real and simulated harmful intent and self-interest attributions. (E) Averaged real (grey) and simulated (coloured) harmful intent and self-interest attribution for each condition across all trials. Analysis of simulated data using a mixed effects model with ID as a random variable suggested pre-existing paranoia was positively associated with harmful intent (0.11, 95%CI: 0.05, 0.16; model S5a) but not self-interest (-0.02, 95%CI: -0.07, 0.03; model S5b), and being paired with an initially unfair Dictator did not influence harmful intent attributions, but led to larger self-interest attributions (0.27, 95%CI: 0.18, 0.37; model S4b; see S6 Fig for comparison with real data across both conditions).
(DOCX)

**S8 Fig. Model comparison for the belief-based social model.** The 1-$\eta_{dg}$ Bayes-Belief model (BB1eta) came first overall across the groups. Each model set was fitted using mixed-effect concurrent Bayesian modelling (Piray et al., 2018) for each group in our population. Model frequency represents the predominance of model $k$ in the population; it is the frequency of times model $k$ best fits all participants. Exceedance probabilities demonstrate the probability that model $k$ is more commonly expressed than any other model in model space. Protected exceedance probabilities are more conservative as they also include the null–that no model best describes the data (Piray et al., 2018). HP = High Paranoia; HI = High ICAR score; LP = Low Paranoia; LI = Low ICAR score.
(DOCX)

**S9 Fig. Partial spearman correlation matrices.** (A) Partial correlations between all social parameters only. (B) Partial correlations between social parameters and tau from the non-social model.
(DOCX)

**S10 Fig. Recovery analysis of the winning social model.** X = non-significant relationship.
(DOCX)

**S11 Fig. Simulated differences of policy and attributions at several wSI values.** (A & B) Initial policy map differences between those with high and low paranoia. Plots were constructed by using the mean w0, wSI, and wHI of those with high (persecutory ideation > 3.66) and low (persecutory ideation < 3.66) paranoid participants within our sample. Mean parameter estimates for low paranoia: w0 = -0.935, wHI = 0.102, wSI = 0.129. Mean parameter estimates for high paranoia: w0 = -1.174, wHI = 0.121, wSI = 0.158. (C) Simulated attributional changes at 10 different values (0–1) of wSI with all other parameters fixed (pHI0 = 0.5, uHI0 = 2, pSI0 = 0.5, uSI0 = 2, uPi = 2, w0 = -1, wHI = 0.1, wSI = 0.1–0.9, $\eta_{dg}$ = 0.5). For each wSI value,

100 synthetic participants were created.
(DOCX)

**S12 Fig. Network analysis between social parameters and paranoia from Barnby et al., 2020.** (A) Our nonparanormal network replicated results from Barnby et al., (2020). (B) Stability analysis demonstrated satisfactory case-dropping estimates. (C) Bootstrapped edge weights demonstrated satisfactory estimates. See S3 Table for all edge statistics in the network.
(DOCX)

**S13 Fig. Isolated network to test collider bias between nodes.** Paranoia is robustly correlated with pHI0 and uπ; the independent relationship between pHI0 and uπ may therefore be at high risk of collider bias.
(DOCX)

**S1 Table. Non-Social Associative Model Statistics.** 'RW' refers to the Rescorla-Wagner (RW) / Q-learning learning model. 'PH' refers to the Pierce-Hall salience model. 'WS' refers to the 'Win-Stay; Lose-Switch' model.
(DOCX)

**S2 Table. Social Model Comparison Statistics.** LL, BIC, and AIC figures are indicative of the summed log probability from the combination of harmful intent and self-interest estimates for each model fitted using Maximum-A-Priori techniques. Bold highlighting represents winning models in each class.
(DOCX)

**S3 Table. Bootstrapped estimates for each edge in the replication network.**
(DOCX)

**S4 Table. Top Model Average of Variables Associated with decision temperature (τ).** All regression estimates are extracted from Model J2c in the analysis code. $w_{SI}$ was not included in the final top model and therefore excluded from this table.
(DOCX)

**S1 Text. Associative Social Model Formalism.**
(DOCX)

## Acknowledgments

We would like to greatly thank Vaughan Bell, Nichola Raihani, and Andreea Diaconescu for their comments which substantially improved the manuscript.

## Author Contributions

**Conceptualization:** Joseph M. Barnby, Mitul A. Mehta.

**Data curation:** Joseph M. Barnby.

**Formal analysis:** Joseph M. Barnby.

**Funding acquisition:** Joseph M. Barnby.

**Investigation:** Joseph M. Barnby.

**Methodology:** Joseph M. Barnby, Michael Moutoussis.

**Project administration:** Joseph M. Barnby.

**Resources:** Joseph M. Barnby.

**Software:** Joseph M. Barnby, Michael Moutoussis.

**Supervision:** Mitul A. Mehta, Michael Moutoussis.

**Validation:** Joseph M. Barnby, Michael Moutoussis.

**Visualization:** Joseph M. Barnby.

**Writing – original draft:** Joseph M. Barnby.

**Writing – review & editing:** Joseph M. Barnby, Mitul A. Mehta, Michael Moutoussis.

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
