## [Decision Letter · Decision Letter 0]

26 May 2022

Dear Dr Barnby,

Thank you very much for submitting your manuscript "The computational relationship between reinforcement learning and social inference in paranoia" for consideration at PLOS Computational Biology.

As with all papers reviewed by the journal, your manuscript was reviewed by members of the editorial board and by several independent reviewers. In light of the reviews (below this email), we would like to invite the resubmission of a significantly-revised version that takes into account the reviewers' comments.

We cannot make any decision about publication until we have seen the revised manuscript and your response to the reviewers' comments. Your revised manuscript is also likely to be sent to reviewers for further evaluation.

Sincerely,

Samuel J. Gershman

Deputy Editor

PLOS Computational Biology

Reviewer's Responses to Questions

**Comments to the Authors:**

Reviewer #1: Summary:

This study aims to compare the relationship between paranoia and both social and non-social inferences in probabilistic learning. Key to their question is whether or not known dysfunction in reversal learning generalizes or is specific to misattribution of harmful intent in social situations. This is an interesting question to assess whether or not paranoia is specific to social interactions or a symptom of a more generalized deficit. The results are a promising step forward in understanding the mechanisms of persecutory ideation, and the discussion suggests several future directions built on this work. However, I found I needed several re-reads to understand the methodology and how it answered the aims of the study.

Major concerns

1. The current organization of the paper makes it difficult to track what is important. An additional paragraph at the beginning of the results section that summarizes the key methodology would greatly help with understanding the results without jumping around the paper to the methods.

2. I am having trouble following the models as the document is currently written. Are there terms that can be applied to the symbols described to increase clarity instead of just symbols? Furthermore, in the discussion, you don’t cite the symbols, which makes it even more confusing to switch from only symbols to only text. Please review the results and discussion section for consistency in language.

3. The use of symbols without defining the meaning is very confusing; even more confusing is that some symbols for parameters never get defined (for example, pi). Figure 2 partially explains the parameters as they fit in the model, but would benefit from clearer labels or text in the figure description.

4. In S4, it seems like the blue and green lines do not map on well to the same cards in the simulation in phase 2, suggesting it is not a great fit for the data. Could you please clarify?

5. From the introduction, abstract, and title, I assumed that the comparison of reinforcement modeling in the non-social task with the social intent parameters was a major result of the paper, but the actual result comparing the two models is brief. Would it be possible to include Figure S9 in the main text to more clearly visualize this finding?

Smaller comments

1. Not all acronyms are defined, making it difficult to keep track of what is important. ICAR is one example. Other examples are HI and SI. I figured out what they mean by context, but these acronyms mean something different to me (homicidal and suicidal ideation), which led to additional confusion. Please define all acronyms.

2. Why is eta sometimes written out and sometimes a symbol? Are they different?

3. I found the abstract was misleading about the sample size and recommend changing it to the size of the final dataset used.

4. Typo pg 27 - “A simulated annealing approach on parameter values was combined was followed by gradient-ascent on MAP to minimize the chance of missing important MAP maxima.”

5. Typo pg 15 - “Larger priors beliefs”

6. Typo pg 14 - “optimal choices are not ignored in those who are more paranoid, optimal choices may instead take longer on average to become exploited, leaving more room for ambiguity.”

7. I found this sentence in the abstract confusing. Consider rewriting: "Consistent with prior work we show that, after reversals, pre-existing paranoia was independently associated with uncertainty around a partner’s behavioural policy and rigidity in harmful intent attributions in the social task, and in the probabilistic task, larger decision temperatures and commitment to suboptimal cards."

8. I appreciated the summary of the findings at the end of sections, for example on the middle of page 8. It may be helpful to implement this elsewhere, as there is a lot of repetition when looking at results with and without confounders that makes it harder to track the key findings.

9. I appreciate the use of preregistration and making your code available.

Reviewer #2:

I read and enjoyed Barnby and colleagues paper. I believe that it should be published.

My concerns should be addressed in a revision, but each is about design and interpretation, rather than a fundamental flaw.

The authors administer their trust task alongside a probabilistic reversal learning task. This is in an attempt to reconcile apparent conflicts between their work and others who favor a non-social interpretation of the mechanisms underlying paranoia.

Interestingly they replicate both their own effect, and the non-social effect - both tasks relate to paranoia in a manner that has been shown previously.

Intriguingly, the authors also report correlations between the key model parameters from each type of task.

This suggests some common mechanism.

The authors prefer to imply that everything is social, rather than appealing to the perhaps more parsimonious suggestion that their results reflect a simpler non-social learning mechanism in play in both their social and non-social tasks.

I am not sure that this approach could have given them the dissociation that they claim. Perhaps if they had shown no relationship between paranoia and reversal learning they would be correct to center the social. As it stands, the result is ambiguous and open to interpretation.

Their all-social interpretation seems to stem from the assymmetry towards the social in their social task. Could it be though that the social task is simply more complex and recursive and more demanding of a general learning mechanism than their non-social reversal task?

I would suggest that they temper the "all social" interpretation, and perhaps look to future studies to disambiguate more decisively. For example, Lockwood et al suggest that a social brain explanation of some phenomenon might be more warranted if the social computations are realized neurally in different structures or circuits: https://www.cell.com/trends/cognitive-sciences/fulltext/S1364-6613(20)30168-6

Recent meta-analysis of social and non-social prediction error studies did identify a set of common and relatively unique regions (including the DMPFC) where social computations might be implemented (https://pubmed.ncbi.nlm.nih.gov/35017672/). It strikes me that these predictions about implementation might be less ambiguous than comparisons of task behaviors.

There are also some minor issues with scholarship and citation. They suggest that Reed et al included a social task, they did not. They fail to acknowledge that Suthaharan et al compared social and non-social Probabilistic reversal learning and failed to find a differential relationship with paranoia.

They could also focus more on studies that have employed tasks in which the social and non-social are pitted against each other in the same task. Diaconescu et al springs to mind. There no specific effect of schizophrenia was found on the social aspect of the task (though no paranoia association was reported, we might imagine that SZ patients would be more paranoid, since 90% of them report paranoia). Also conspicuous in its absence is a treatment of Rossi-Goldthorpe et al - where a manipulation of social group identity changed prior beliefs but did not interact with paranoia - instead paranoia was more strongly related to non-social aspects of the task and computational model (paranoid participants inferred that their own task performance was unreliable).

I also feel its a little inaccurate and overly simplistic to appeal to glutamate mediating priors and dopamine relatng to social processing.

The authors might consider these this paper when unpacking their results in the context of neurochemistry:

https://pubmed.ncbi.nlm.nih.gov/35289748/

These data suggest that the form of what is being learned and the contingencies (whether social or non-social) are crucial to the impact of dopamine manipulations.

**Have the authors made all data and (if applicable) computational code underlying the findings in their manuscript fully available?**

Reviewer #1: Yes

PLOS authors have the option to publish the peer review history of their article (what does this mean?). If published, this will include your full peer review and any attached files.

Reviewer #1: No
---

## [Decision Letter · Decision Letter 1]

23 Jun 2022

Dear Dr Barnby,

We are pleased to inform you that your manuscript 'The computational relationship between reinforcement learning, social inference, and paranoia' has been provisionally accepted for publication in PLOS Computational Biology.

Best regards,

Samuel J. Gershman

Deputy Editor

PLOS Computational Biology

Reviewer's Responses to Questions

**Comments to the Authors:**

Reviewer #1: Thank you for addressing my concerns regarding the presentation of the manuscript. The changes have greatly increased the understanding of your results.

The addition of the new figure 3 allows for readers to visualize the skewed spread of the paranoia measurement. I think you have appropriately controlled for this skew using a Spearman test, but it is still worth noting that it is an uneven sampling. You have adequately brought up hypotheses and limitations regarding this potential issue in your discussion.

Given your responses to myself and the other reviewer, I have no additional concerns regarding the manuscript.

**Have the authors made all data and (if applicable) computational code underlying the findings in their manuscript fully available?**

Reviewer #1: Yes

PLOS authors have the option to publish the peer review history of their article (what does this mean?). If published, this will include your full peer review and any attached files.

Reviewer #1: **Yes: **Rebecca Kazinka

---

## [Editor Report · Acceptance letter]

14 Jul 2022

PCOMPBIOL-D-22-00450R1 

The computational relationship between reinforcement learning, social inference, and paranoia

Dear Dr Barnby,

I am pleased to inform you that your manuscript has been formally accepted for publication in PLOS Computational Biology. Your manuscript is now with our production department and you will be notified of the publication date in due course.

With kind regards,

Zsofia Freund
